# Shrinkage and Durability Evaluation of Environmental Load-Reducing FRPCM by Using Silicone Oil

**DOI:** 10.3390/ma12081240

**Published:** 2019-04-15

**Authors:** Hyeonggil Choi, Heesup Choi, Bokyeong Lee, Dong-Eun Lee

**Affiliations:** 1School of Architecture, Kyungpook National University, 80 Daehakro, Bukgu, Daegu 41566, Korea; hgchoi@knu.ac.kr; 2Department of Civil and Environmental Engineering, Kitami Institute of Technology, 165 Koencho, Kitami, Hokkaido 090-8507, Japan; 3Intelligent Construction Automation Center, Kyungpook National University, 80 Daehakro, Bukgu, Daegu 41566, Korea; bklee11@knu.ac.kr; 4School of Architecture and Civil Engineering, Kyungpook National University, 80 Daehakro, Bukgu, Daegu 41566, Korea; dolee@knu.ac.kr

**Keywords:** FRPCM, silicone oil, expansive additive, shrinkage reducing agent, shrinkage, durability

## Abstract

In this study, the shrinkage and durability of environmental load-reducing fiber-reinforced polymer cement mortar (FRPCM) were examined by using silicone oil. The results indicated that the shrinkage can be greatly reduced by adding silicone oil. However, when the silicone oil is added in excess, it affects the strength and durability. Therefore, it is possible to obtain the ECO-FRPCM which gives the effect of reducing the shrinkage and maintaining the strength and durability by adding 3% of silicone oil. From the viewpoint of shrinkage reduction, the use of silicone oil is effective as a substitute for an expansive additive or shrinkage reducing agent used in ECO-FRPCM. Also, by appropriately using silicone oil in combination with an expansive additive or a shrinkage reducing agent, shrinkage can be effectively reduced as compared with the conventional ECO-FRPCM. However, when the silicone oil and the shrinkage reducing agent are used in combination, the shrinkage cannot be efficiently reduced. It is considered that the combination of silicone oil and shrinkage reducing agent influences the mutual shrinkage reduction mechanism, but this needs to be further examined in the future.

## 1. Introduction

Since concrete cracks due to shrinkage significantly influence the safety and durability of concrete structures, it is important to identify them [1,2,3]. Particularly as requirements for long life and high durability of concrete structures have increased in recent years, it is very important to repair these cracks or deteriorated parts efficiently [1,2,3]. As one of the typical repair materials, fiber-reinforced polymer cement mortar (FRPCM) has been researched and developed, and is now commercialized and used as a well-known repair material for cracks or deterioration [4,5]. FRPCM is a material that can block the deterioration factors by efficiently repairing cracks through the crack control and toughness improvement effect through the fibers and excellent adhesion due to the polymer [4,5,6].

In the meantime, environmental-load-reducing FRPCM (ECO-FRPCM), in which ground-granulated blast furnace slag (an industrial by-product) is added as a binder to reduce the environmental loads, has been developed in recent years [7,8]. However, when ground-granulated blast furnace slag is used as a binder, it causes a problem in strength development and an increase in shrinkage at the early curing stage and acceleration of concrete carbonation in the long term [9]. 

Moreover, it has been reported that the silicone oil is used for a cementitious material, thereby improving the carbonation and drying shrinkage resistance due to the water repellency of the silicone oil [10,11,12,13].

Thus, this study investigated the shrinkage characteristics, strength characteristics, and durability with addition of a silicone oil, which was found to be excellent in shrinkage reduction effect or resistance against carbonation, to the FRPCM to overcome the drawback of ECO-FRPCM (with added ground-granulated blast furnace slag). The study also investigated the shrinkage characteristics of ECO-FRPCM where a silicone oil was added (as a substitute of an expansive additive or shrinkage-reducing agent) that was known to have the shrinkage reduction effect.

## 2. Experimental Program

### 2.1. Outline of Experiment

Table 1, Table 2 and Table 3 shows the materials used, binding materials properties and experimental compositions. The ECO-FRPCM used in this experiment employed ground-granulated blast furnace slag as fine aggregates and binder, and expansive additive and shrinkage-reducing agent were added as shrinkage reducing materials. Experiments were conducted by referring to the mix design of ECO-FRPCM, in which the content ratio of the ground-granulated blast furnace slag was set to 35%, silica fume was set to 5%, polymer set to 2% (B × wt.%), water-cement ratio was 36%, slag-to-binder ratio (S/B) was 1, and 0.5 vol.% of polypropylene fiber (6 mm) was admixed [7,8]. Further, antifoaming agent, air entraining agent, and water reducing agent were added to control the flash property. In Series I, the effects of the addition of the silicone oil on the performance of ECO-FRPCM were investigated. The addition rate of the silicone oil was set to five levels (0%, 1%, 3%, 5%, and 10%), and slump flows, strength characteristics, cure shrinkage, carbonation, and freeze-thaw resistance were evaluated for each. In Series II, the effects of the combining use of silicone oil-added ECO-FRPCM with the expansive additive and shrinkage-reducing agent on the strength and shrinkage characteristics were evaluated to investigate the effect of substitution of the expansive additive and shrinkage reducing agent by the silicone oil. The specimens were made by fixing the silicone oil addition to 3% while controlling the addition of the expansive additive and shrinkage reducing agent to have four types of specimens in the experiment. Mixing of FRPCM was performed that all materials put in the pan type mixer excluding antifoaming agent, and then the pre-mixing was performed for 30 s and main mixing was conducted for 3 min. Afterwards, an antifoaming agent was placed in the mixer and the mixing was performed for 1 additional minute. After confirming that the fibers were uniformly mixed, a specimen was prepared. 

### 2.2. Experiment Method

Slump flow immediately after mixing and air content were measured for the properties of the fresh concrete [14,15]. The specimens for the strength test were Ø50 mm × 100 mm cylindrical specimens, which were demolded at an age of 1 day and cured in the constant temperature and humidity chamber at 20 °C and 60% relative humidity. Then, the compressive strength was measured at 3, 7, 28, and 91 days according to JIS A1108 [16]. The tensile strength was measured as follows: a dumbbell-shaped thin specimen was fabricated as shown in Figure 1 and cured under the same environment as that of the compressive strength specimens to have a constant strain by controlling loading rate at 0.5 mm per sec. The direct tensile strength was then measured at 28 days using the universal tester [17].

The length change experiment was conducted by referring to the NEXCO test method 432 [18], which is a curing shrinkage test method of mortar for spray-coating to repair a cross-section, to fabricate a prism specimen (40 mm × 40 mm × 160 mm). This was then demolded at two days and cured in a constant temperature and humidity chamber at 20 °C and 60% relative humidity. In the experiment to measure length-change of curing shrinkage and mass-change rate was conducted in accordance with JIS A 1129-3 [19] by shifting the dial gage method up to the pre-determined age.

In the meantime, the curing shrinkage experiment of NEXCO test method 432 [18] did not consider the shrinkage and expansion properties until the demolding time (at two days). Thus, it was particularly important to identify the expansion phenomenon at the early stage if the expansive additive was used. In this regard, the expansion rate up to two days after fabrication was measured in the sealed curing condition at 20 °C by attaching a 10-mm-long strain gage to the Ø100 mm × 200 mm cylindrical form (summit mold), as shown in Figure 2, in accordance with JIS A 009-2012 [20] to verify the expansion rate at the early stage.

The accelerated carbonation test was conducted to measure the carbonation resistance of concrete using a prism specimen (40 mm × 40 mm × 160 mm) in accordance with JIS A 1153 [21]. The accelerated conditions were 20 ± 2 °C, 60 ± 5% of humidity, and 5 ± 0.2% of CO_2_ concentration. The freezing and thawing test was conducted in accordance with JIS A 1148 Procedure A (Rapid Freezing and Thawing in Water) [22] to set the freezing temperature to −18 °C and the thawing temperature to 5 °C. One cycle was set to 3 h, and measurements were conducted for up to 300 cycles and the relative dynamic elastic modulus was measured every 30 cycles.

## 3. Experiment Result and Discussion

### 3.1. Study on the Effect of Silicone Oil on the Performance of Environmental-Load-Eeducing FRPCM (Series I)

#### 3.1.1. Fresh Property and Strength Characteristics

Figure 3 shows the slump flow and air content results as fresh concrete properties. Since SO0 employed an antifoaming agent to control the air content, its air content was relatively small. However, as the silicone oil addition increased, the slump flow and air content tended to decrease slightly overall. This was because the silicone oil was added as an inner percentage to the mixed water equal to the chemical admixture, so the mixed water amount was reduced as the addition of silicone oil increased.

Figure 4 shows the test results of compressive strength. As the silicone oil was added, the compressive strength was degraded. Thus, the compressive strength of SO10 was reduced by 40% of that of SO0 approximately at the age of 91 days, indicating that care should be taken to the reduction in strength when too much silicone oil was added.

For the tensile strength, only the results from fractured specimens within the target fracture point (center 80 mm) were summarized, which are shown in Figure 5. The tensile strength tended to decrease the same amount as the compressive strength as the silicone oil was added.

The scanning electron microscope observation was conducted in the fracture surface of SO0 and SO10 to investigate the cause of the strength reduction. The results are presented in Table 4. SO10 verified that the fiber was pulled out while maintaining its shape as the adhesion between the fiber and matrix was degraded due to the silicone oil. Compared to the above phenomenon in SO10, the adhesion between the fiber and matrix in SO0, which did not add silicone oil, was strong, so that the trace of resistance to the matrix around the fiber was verified. Thus, the reduction in adhesion between the fiber and matrix due to the addition of silicone oil was considered to have an impact on the strength characteristics, although additional investigation on this is needed.

#### 3.1.2. Shrinkage and Durability Properties

The results of length and mass change rates up until the age of 91 days are shown in Figure 6 and Figure 7. As verified in Figure 6, as the silicon oil content increased, the shrinkage was significantly degraded, with the shrinkage of SO10 being reduced more by 705 × 10^−6^ (about 58%) compared to that of SO0 at the age of 91 days. In the meantime, the mass change rate verified that the length change rate of the silicone oil-added specimen was smaller than that of the specimen without the addition of silicone oil at an early age, but those of SO5 and SO10 were similar with that of SO0 as the age passed, which did not verify a significant correlation. Generally, the shrinkage reducing mechanism occurs due to the reduction in surface tension of pore solution or changes in pore structure. Thus, as a pore diameter is smaller, the capillary tension becomes larger, thereby increasing the shrinkage [23]. When pores whose diameter is different are adjacent, moisture inside the pores is moved to a smaller-diameter pore due to capillarity. However, if silicone oil, which is water repellent, is added, the pore wall will have the water repellent property, resulting in difficulties in moisture travel to a smaller diameter pore, but easy moisture travel to a larger diameter pore [13,24]. Accordingly, as the capillary tension inside the pore was smaller, shrinkage was likely to be reduced significantly. Thus, although there was no significant correlation between general mass change rate and shrinkage, the change in moisture travel inside the matrix influenced the shrinkage phenomenon according to the addition of silicone oil.

The accelerated carbonation test results are presented in Table 5. The progress of carbonation is a reaction that is significantly affected by the moisture condition inside the material [25]. A previous study reported that the water repellent effect caused by the silicone oil reduced the adsorption water inside the material, thereby restraining carbonation [24]. However, carbonation was not verified at the age of eight weeks in the accelerated carbonation test. The carbonation effect according to the presence of silicone oil could not be verified as the carbonation of ECO-FRPCM was difficult inherently due to the fiber reinforcement, but the carbonation suppression effect was expected in the long-term age.

The freeze-thaw test results are shown in Figure 8. SO0, to which silicone oil was not added, maintained 100% of the relative dynamic elastic modulus even at the time when 300 cycles were passed whereas the relative dynamic elastic modulus of silicone oil-added SO3 or SO5 specimen tended to decrease somewhat. The previous studies [13,24] explained that the water pressure reducing mechanism due to the entrained air was not working, which was inherently expected, as the movement of non-frozen water was reduced due to the silicone oil when the moisture inside the capillary pore was frozen because silicone oil is present as a form of oil droplet inside the pore if the silicone oil is added. The same tendency could also be verified in ECO-FRPCM.

As described above, when silicone oil was added to ECO-FRPCM, shrinkage could be reduced significantly as the addition increased from the viewpoint of shrinkage reduction. However, the strength or durability was degraded if excessive silicone oil was added. Considering this, the proper amount of silicone oil addition was regarded as around 3%. Based on this result, the silicone oil addition was set to 3% constantly in Series II and the effect of silicone oil as a replacement of expansive additive or shrinkage reducing agent on the strength and shrinkage characteristics of ECO-FRPCM was evaluated.

### 3.2. Investigation of Silicone oil Effect as a Replacement of Expansive Additive and Shrinkage Reducing Agent (Series II)

#### 3.2.1. Fresh Property and Strength Characteristics

Figure 9 shows the results of the fresh concrete properties. The figure verified that as the silicone oil was added, a slump flow had no significant change and the air content tended to decrease slightly, but the result satisfied the range of target air content (6.0 ± 1.5%).

Figure 10 shows the results of the compressive strength test. The compressive strength of the specimen with the addition of silicone oil was reduced slightly at the age of 91 days, but the difference was minimal.

Thus, the silicone oil as a replacement of expansive additive or shrinkage reducing agent used in ECO-FRPCM for the effect of shrinkage reduction should have no significant problem in constructability and strength development if the addition was in a range of 3%.

#### 3.2.2. Shrinkage Properties

The expansion rate test was conducted with specimens mixed with expansive additive to verify the expansion effect at an early age when the expansive additive was mixed. The results are shown in Figure 11. As the reference results, Figure 11 also shows the results of SO0 to which the expansive additive was not added. The figure verified the expansion rate at an early age according to the expansive additive content, and specimens where the expansive additive was not mixed did not show the expansion.

Figure 12 shows the results of the length change results in the cure shrinkage test. At the age of 28 days, the specimens of ES1/2-SO3 and E-SO3 showed a smaller value than 500 × 10^−6^, which was the reference value of dimensional stability of repair material, which satisfied the reference value, but ES where silicone oil was not added, and S-SO3 where the shrinkage reducing agent and silicone oil were both added, did not satisfy the reference value. In addition, the shrinkage of ES1/2-SO3 where the expansive additive, shrinkage reducing agent, and silicone oil were added was approximately 750 × 10^−6^ at the age of 91 days, which verified that the effect of shrinkage reduction was the largest under these conditions. The length change results considering the early expansion phenomenon is shown in Figure 13. The figure verified that the ES specimen, which was a mix of ECO-FRPCM, satisfied the reference value at the age of 28 days by considering the early expansion phenomenon, and all the ES1/2-SO3, E-SO3, and ES specimens had a shrinkage around 600 × 10^−6^ at the age of 91 days. However, the shrinkage of S-SO3, to which shrinkage reducing agent and silicone oil were added, increased at a significant rate at all ages compared to other levels. Thus, a difference of more than 400 × 10^−6^ was exhibited compared to other specimens at the age of 91 days.

In the meantime, the effects of shrinkage reduction by silicone oil were investigated summarizing the author’s previous study results [8] and the experimental results. The length change rate and shrinkage reduction results at the age of 28 days are shown in Figure 14. The SO0 and SO3 specimens, to which silicone oil was added singly, and E-SO0 and E-SO3 specimens, to which both of silicone oil and expansive additive were added, had the shrinkage reduction effect of 14.1% and 44.8%, respectively whereas the shrinkage of S-SO0 and S-SO3 specimens, to which both of silicone oil and shrinkage reducing agent were added, tended to rather increase by 23.8%. This was due to the difference in the shrinkage reducing mechanism between silicone oil and shrinkage reducing agent. That is, the shrinkage occurs due to the change in the surface tension of pore solution generally as shown in Figure 15, and the smaller the pore diameter was, the larger the shrinkage was as the capillary tension increased (a). In the meantime, silicone oil exhibited the shrinkage reducing effect through the water-repellent performance (b), and the shrinkage reducing agent reduced the surface tension of the pore solution inside the capillary pore thereby exhibiting the shrinkage reducing effect (c). Here, when both the silicone oil and shrinkage reducing agent were added, the interfacial activation action via the shrinkage reducing agent alleviated the water-repellent property of silicone oil as well as changing the surface tension inside the pore and water movement inside the pore of the hardened body, thereby affecting the mutual shrinkage reducing mechanism between silicone oil and the shrinkage reducing agent, resulting in no efficient shrinkage reduction. However, this needs to be studied in the future.

## 4. Conclusions

This study investigated the shrinkage characteristics and durability of silicone oil-added environmental-load-reducing FRPCM, and the following conclusions were obtained.
(1)A shrinkage reduction effect at a significant rate could be obtained as the silicone oil content increased. However, excessive addition may affect the strength and durability. Thus, approximately 3% of silicone oil content was appropriate to develop strength and maintain durability while obtaining ECO-FRPCM that had the shrinkage reduction effect.(2)The use of silicone oil as a replacement of expansive additive or shrinkage-reducing agent used in ECO-FRPCM was effective in terms of shrinkage reduction. In particular, equivalent or better shrinkage-reducing effect could be obtained by combining both silicone oil and expansive additive, or by combining silicone oil, expansive additive, and shrinkage-reducing agent.(3)When silicone oil and the shrinkage reducing agent were combined, the inherent shrinkage-reduction effects of each were difficult to distinguish. This may affect each of the shrinkage-reducing mechanisms, such as alleviating not only the water-repellent property of silicone oil, but also moisture travel inside the pores of the hardened body and changes in surface tension, resulting in inefficient shrinkage reduction. However, this finding requires additional study in the future.

## Figures and Tables

**Figure 1 materials-12-01240-f001:**
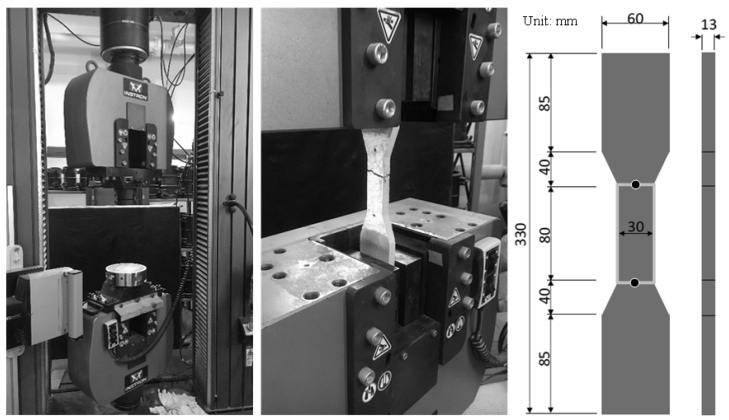
Tensile strength testing machine and size of the specimen.

**Figure 2 materials-12-01240-f002:**
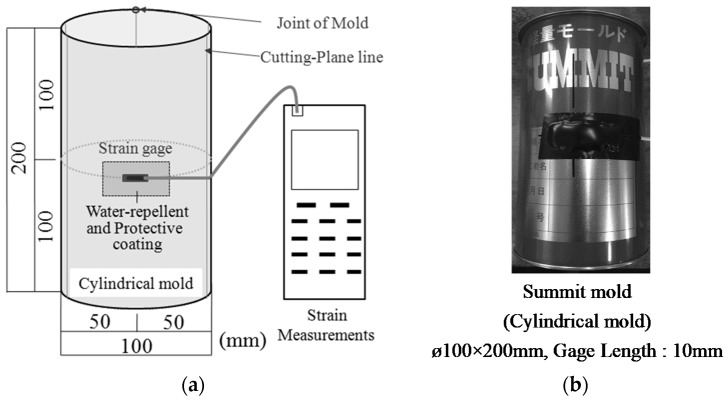
Overview of expansion rate test [20]. (**a**) Outline of the measurement of expansion rate; (**b**) Summit mold.

**Figure 3 materials-12-01240-f003:**
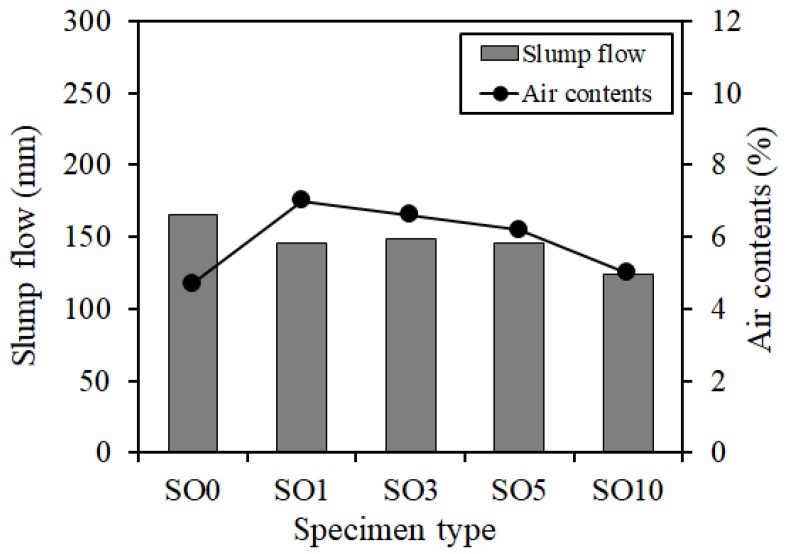
Fresh properties.

**Figure 4 materials-12-01240-f004:**
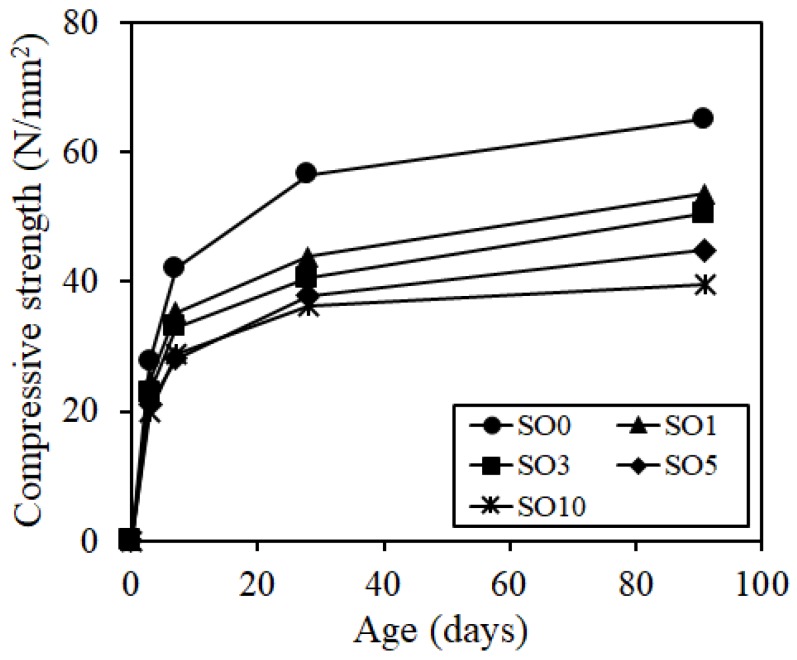
Compressive strength.

**Figure 5 materials-12-01240-f005:**
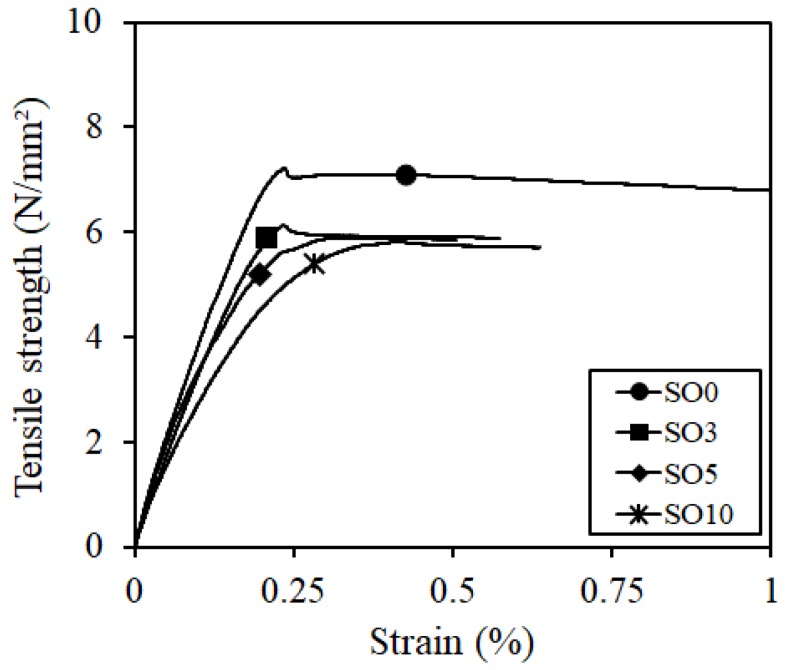
Tensile strength.

**Figure 6 materials-12-01240-f006:**
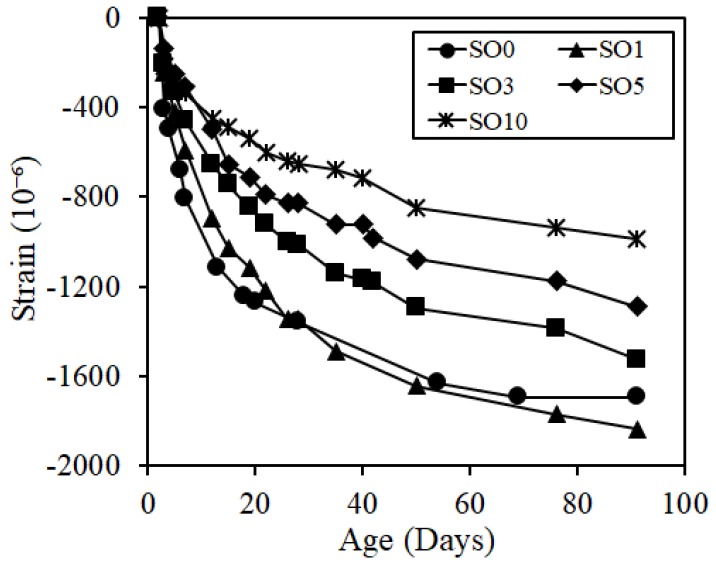
Cure shrinkage.

**Figure 7 materials-12-01240-f007:**
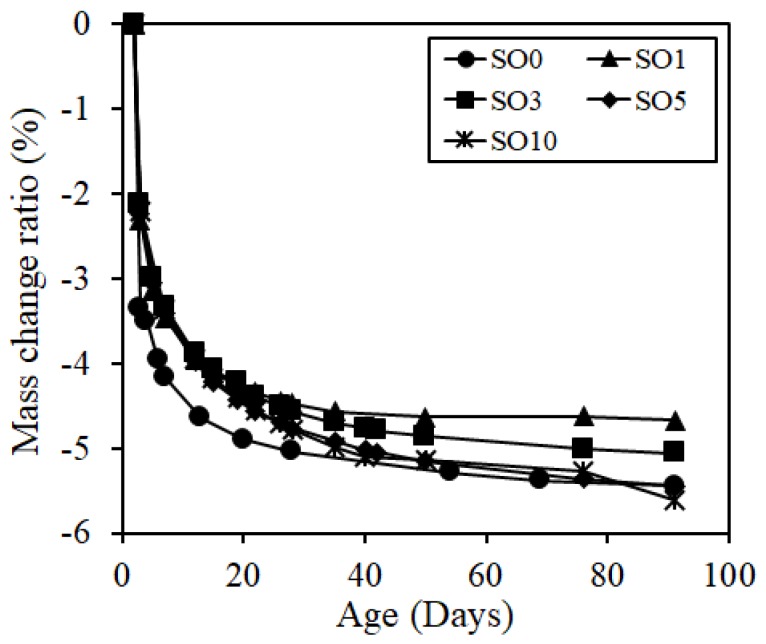
Mass change ratio.

**Figure 8 materials-12-01240-f008:**
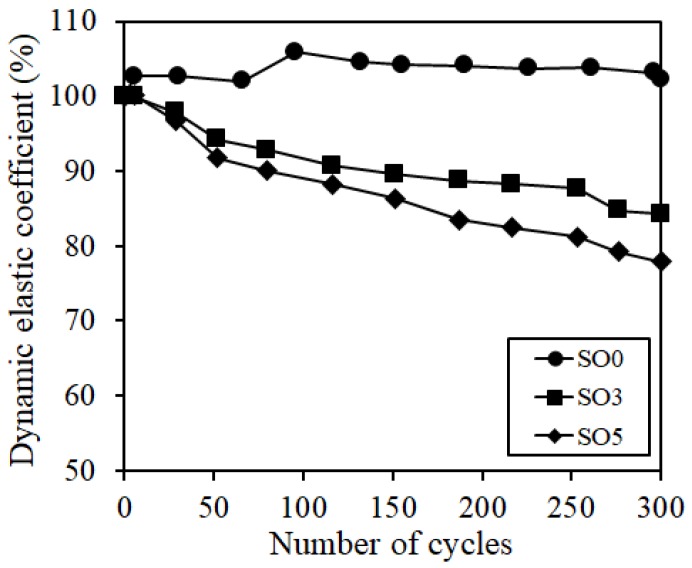
Dynamic elastic coefficient.

**Figure 9 materials-12-01240-f009:**
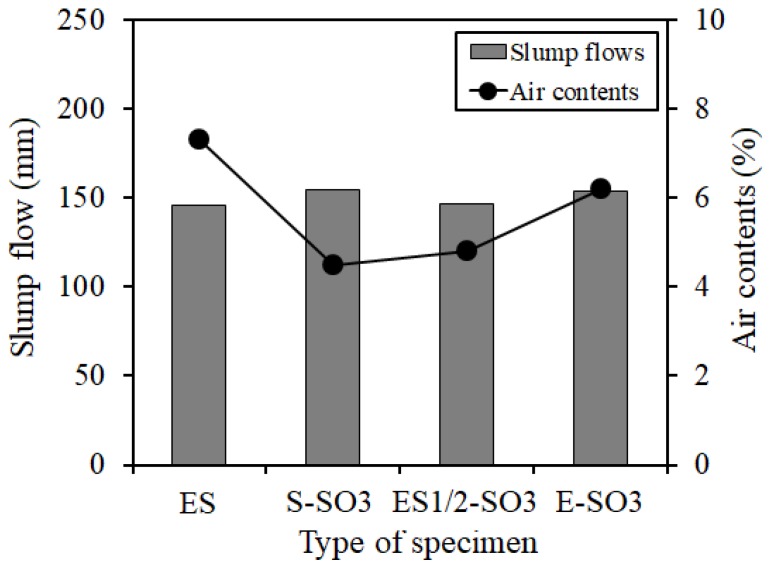
Fresh properties.

**Figure 10 materials-12-01240-f010:**
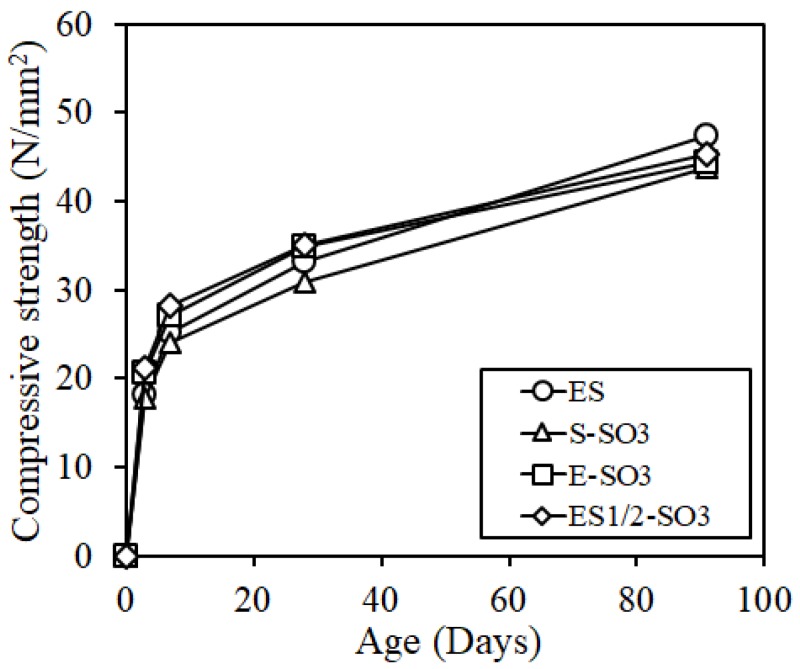
Compressive strength.

**Figure 11 materials-12-01240-f011:**
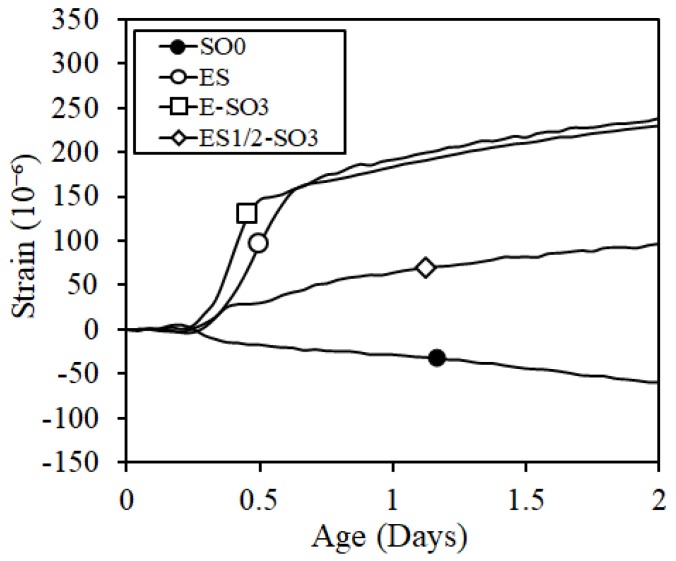
Expansion rate.

**Figure 12 materials-12-01240-f012:**
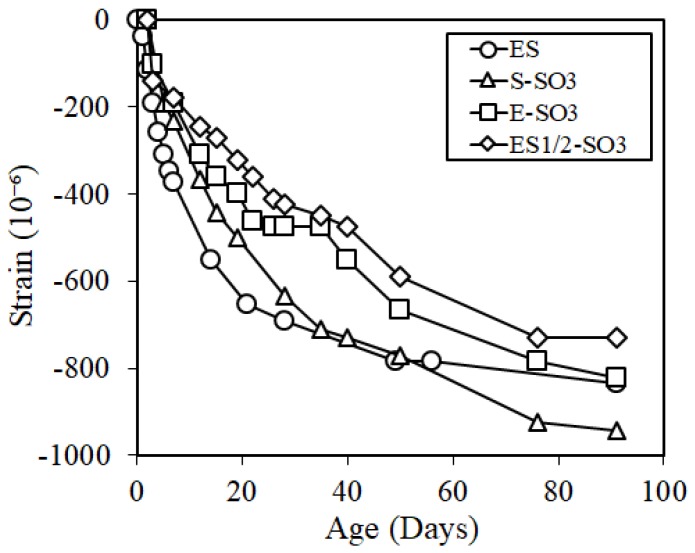
Cure shrinkage.

**Figure 13 materials-12-01240-f013:**
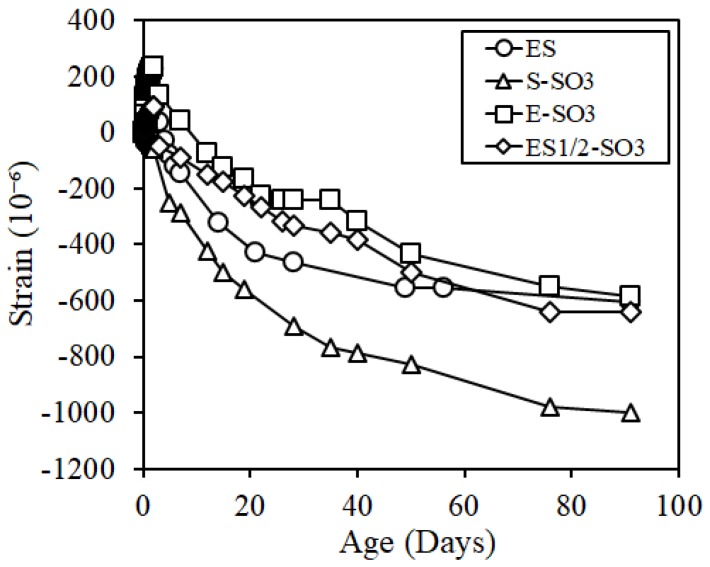
Cure shrinkage considering the early expansion.

**Figure 14 materials-12-01240-f014:**
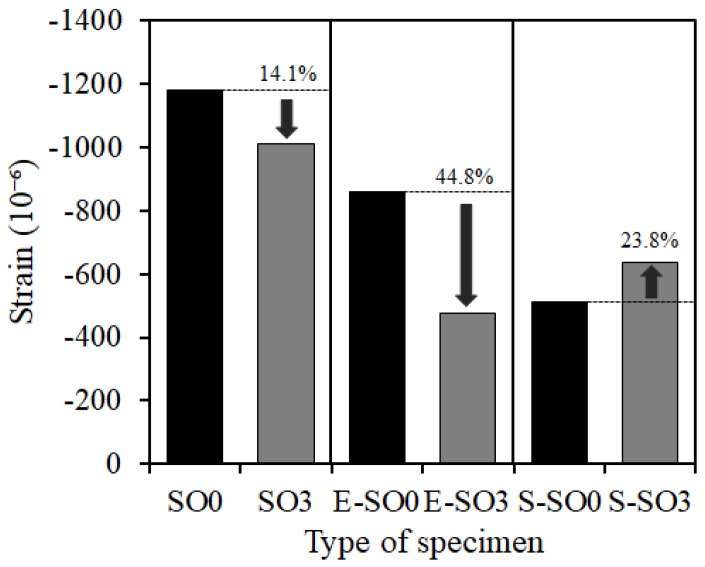
Shrinkage reduction effect of silicon oil in ECO-FRPCM.

**Figure 15 materials-12-01240-f015:**
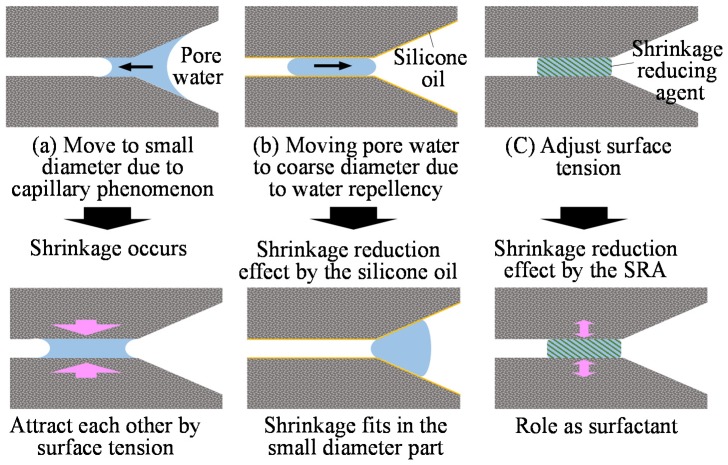
Shrinkage reduction effect.

**Table 1 materials-12-01240-t001:** The materials Used.

Type	
Binder	Ordinary Portland cement (OPC), Density: 3.17 g/cm^3^
	Blast furnace slag (BFS), Blaine: 4000 cm^2^/g, Density: 2.89 g/cm^3^
	Silica fume (SF), Density: 2.25 g/cm^3^
	Expansive additive (E), Ettringite type, Density: 2.93 g/cm^3^
sand	Blast furnace slag sand, Density: 2.66 g/cm^3^, Grain size: 0.6~1.2mm
Admixture	Polymer (P), Acetic acid acrylic type, Density: 1.05 g/cm^3^
	Shrinkage reducing agent (SRA), Powder type nonionic mixture
	Antifoaming agent (AA), Poly ether type
	Air entraining agent (AE), Aliphatic alcohol type, Density: 1.04~1.08 g/cm^3^, Total alkali amount (%): 3.7
	Water reducing agent (WRA), Poly carboxylic type
	Silicone oil (SO), Density: 0.96 g/cm^3^, Viscosity (25 °C): 20cst, Refractive index: 1.42
Fiber	Polypropylene (PP): Length: 6 mm, Diameter: 42.6 μm, Density: 0.91 g/cm^3^

**Table 2 materials-12-01240-t002:** Binding materials properties.

Binder	Specific Surface Area (g/cm^3^)	Density (g/cm^3^)	Chemical Composition (%)							
			SiO_2_	Al_2_O_3_	Fe_2_O_3_	CaO	MgO	SO_3_	f-CaO	Ig.loss
OPC	3500	3.17	21.4	5.5	2.8	64.3	2.1	1.9	0.25	0.56
BFS	4000	2.89	34.0	14.4	0.83	43.3	6.5	-	-	0.1
E	3260	2.93	21.0	5.2	0.8	70.6	-	18.5	49.8	1.58
SF	-	2.25	96.50	0.46	0.13	0.37	0.37	-	-	1.83

**Table 3 materials-12-01240-t003:** Experimental compositions.

Series	Specimen	W/B	S/B	Fiber (Vol.%)	Binder (%)				Admixture (B × wt.%)						Curing Condition	Test Items
					OPC	BFS	SF	E	P	WRA	AA	AE	SO	SRA		
I	SO0	36	1	0.5	60	35	5	0	2	0.07	0.18	0.5	0	0	20 °C, 60% RH	Slump flow. Air contents. Compressive strength. Tensile strength. Cure shrinkage. Carbonation. Freeze-thaw test
	SO1												1			
	SO3												3			
	SO5												5			
	SO10												10			
II	ES	36	1	0.5	52	35	5	8	2	0.07	0.18	0.5	0	4	20 °C, 60% RH	Slump flow. Air contents. Compressive strength. Cure shrinkage. Expansion rate
	ES1/2-SO3				56			4					3	2		
	E-SO3				52			8						0		
	S-SO3				60			0						4		

**Table 4 materials-12-01240-t004:** Scanning electron microscope observation.

Type	SO0	SO10
Shape of surface	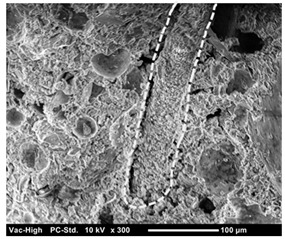	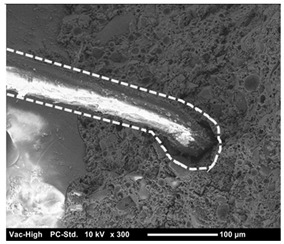
	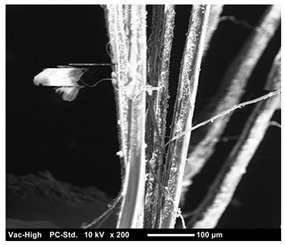	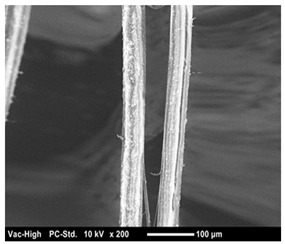

**Table 5 materials-12-01240-t005:** Carbonation depth.

Type	SO0	SO1	S03	S05
4 weeks	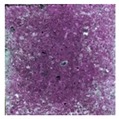	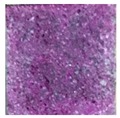	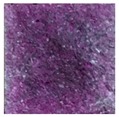	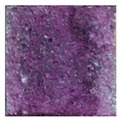
8 weeks	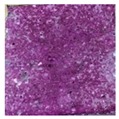	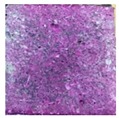	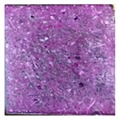	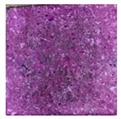

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
