# Peer review of "Shrinkage and Durability Evaluation of Environmental Load-Reducing FRPCM by Using Silicone Oil"

_materials, 2019, doi:10.3390/ma12081240_

Reviewer 1 Report

This is an interesting article regarding current problems related to materials for the repair of  concrete structures.

The research was planned and presented in a way that allows obtaining systematic information on the impact of

silicone oil on the properties of repair cement mortars. The obtained results are interesting and their discussion

is sufficient. Some additions are necessary:

1. in the introduction, reference should be made to existing research in the use of silicone oil as an addition to mortars,

2. it is necessary to provide more detailed information about the materials used and the mortar composition, 3. if possible, it would be beneficial to provide information on the impact of silicone oil on the adhesion of

repair mortars;

if such tests have not been carried out, the reviewer suggests that they should be carried out in further stages

of the research.

Author Response

We would like to thank the reviewer for carefully reading and for giving quite valuable comments and suggestions, which

substantially helped improving the quality of this manuscript. We describe our response point by point to each comment

(in bold letters) and we have marked the modifications in red in the paper. Please refer to the attached file.

Reviewer 2 Report

Comments are in the attached file.

Author Response

(The authors gave the same response as above.)

Reviewer 3 Report

Designing civil engineering materials with a limited impact on the environment is one of the major challenge the sector

must address. The development of environmental load-reducing fibre-reinforced polymer cement mortar (FRPCM) is

then following this necessary trend.

This manuscript presents a study about the shrinkage and durability of FRPCM by using silicone oil instead of classical

shrinkage reducing agent comparing the fresh properties, strength characteristics, the shrinkage evolution and the

carbonation resistance tests. Globally the article is presented correctly, the introduction gives a quick background about

the subject, the methods are clearly explained and the results are correctly discussed, supporting the conclusions. Although

the subject is relatively specific, this manuscript may find its readership in Materials journal.

Some points may be addressed before an eventual publication:

- the introduction is very brief and only gives 9 references. It may be interesting to give more varied references to other repairing

materials for example

- Table 1 and Table 2 may be inverted

- Table 2: some materials are not presented in the text. Please explain the mix and the mixing process more thoroughly

- Table correct Blaine

- l 96 – 98: please explain whether air entraining admixture is included in S01-S010 mixes. AE may be added in table 1

- l 125: please use percentages along with absolute values

- l 169: correct flow

- l 181 – 184: please give a comment on the small extension of ES1/2-SO3

- figure 14: please complete the caption with something like ‘of silicon oil in ECO-FRPCM’

Author Response

(The authors gave the same response as above.)
